# Learning Multiagent Communication
# with Backpropagation

**Sainbayar Sukhbaatar**
Dept. of Computer Science
Courant Institute, New York University
sainbar@cs.nyu.edu

**Arthur Szlam**
Facebook AI Research
New York
aszlam@fb.com

**Rob Fergus**
Facebook AI Research
New York
robfergus@fb.com

## Abstract

Many tasks in AI require the collaboration of multiple agents. Typically, the communication protocol between agents is manually specified and not altered during training. In this paper we explore a simple neural model, called CommNet, that uses continuous communication for fully cooperative tasks. The model consists of multiple agents and the communication between them is learned alongside their policy. We apply this model to a diverse set of tasks, demonstrating the ability of the agents to learn to communicate amongst themselves, yielding improved performance over non-communicative agents and baselines. In some cases, it is possible to interpret the language devised by the agents, revealing simple but effective strategies for solving the task at hand.

## 1 Introduction

Communication is a fundamental aspect of intelligence, enabling agents to behave as a group, rather than a collection of individuals. It is vital for performing complex tasks in real-world environments where each actor has limited capabilities and/or visibility of the world. Practical examples include elevator control [3] and sensor networks [5]; communication is also important for success in robot soccer [25]. In any partially observed environment, the communication between agents is vital to coordinate the behavior of each individual. While the model controlling each agent is typically learned via reinforcement learning [1, 28], the specification and format of the communication is usually pre-determined. For example, in robot soccer, the bots are designed to communicate at each time step their position and proximity to the ball.

In this work, we propose a model where cooperating agents learn to communicate amongst themselves before taking actions. Each agent is controlled by a deep feed-forward network, which additionally has access to a communication channel carrying a continuous vector. Through this channel, they receive the summed transmissions of other agents. However, what each agent transmits on the channel is not specified a-priori, being learned instead. Because the communication is continuous, the model can be trained via back-propagation, and thus can be combined with standard single agent RL algorithms or supervised learning. The model is simple and versatile. This allows it to be applied to a wide range of problems involving partial visibility of the environment, where the agents learn a task-specific communication that aids performance. In addition, the model allows dynamic variation at run time in both the number and type of agents, which is important in applications such as communication between moving cars.

We consider the setting where we have $J$ agents, all cooperating to maximize reward $R$ in some environment. We make the simplifying assumption of full cooperation between agents, thus each agent receives $R$ independent of their contribution. In this setting, there is no difference between each agent having its own controller, or viewing them as pieces of a larger model controlling all agents. Taking the latter perspective, our controller is a large feed-forward neural network that maps inputs for all agents to their actions, each agent occupying a subset of units. A specific connectivity

structure between layers (a) instantiates the broadcast communication channel between agents and (b) propagates the agent state.

We explore this model on a range of tasks. In some, supervision is provided for each action while for others it is given sporadically. In the former case, the controller for each agent is trained by backpropagating the error signal through the connectivity structure of the model, enabling the agents to learn how to communicate amongst themselves to maximize the objective. In the latter case, reinforcement learning must be used as an additional outer loop to provide a training signal at each time step (see the supplementary material for details).

## 2 Communication Model

We now describe the model used to compute the distribution over actions $p(\mathbf{a}(t)|\mathbf{s}(t), \theta)$ at a given time $t$ (omitting the time index for brevity). Let $s_j$ be the $j$th agent's view of the state of the environment. The input to the controller is the concatenation of all state-views $\mathbf{s} = \{s_1, ..., s_J\}$, and the controller $\Phi$ is a mapping $\mathbf{a} = \Phi(\mathbf{s})$, where the output $\mathbf{a}$ is a concatenation of discrete actions $\mathbf{a} = \{a_1, ..., a_J\}$ for each agent. Note that this single controller $\Phi$ encompasses the individual controllers for each agents, as well as the communication between agents.

### 2.1 Controller Structure

We now detail our architecture for $\Phi$ that is built from modules $f^i$, which take the form of multilayer neural networks. Here $i \in \{0, .., K\}$, where $K$ is the number of communication steps in the network.

Each $f^i$ takes two input vectors for each agent $j$: the hidden state $h_j^i$ and the communication $c_j^i$, and outputs a vector $h_j^{i+1}$. The main body of the model then takes as input the concatenated vectors $\mathbf{h}^0 = [h_1^0, h_2^0, ..., h_J^0]$, and computes:

$$h_j^{i+1} = f^i(h_j^i, c_j^i) \tag{1}$$

$$c_j^{i+1} = \frac{1}{J-1} \sum_{j' \neq j} h_{j'}^{i+1}. \tag{2}$$

In the case that $f^i$ is a single linear layer followed by a non-linearity $\sigma$, we have: $h_j^{i+1} = \sigma(H^i h_j^i + C^i c_j^i)$ and the model can be viewed as a feedforward network with layers $\mathbf{h}^{i+1} = \sigma(T^i \mathbf{h}^i)$ where $\mathbf{h}^i$ is the concatenation of all $h_j^i$ and $T^i$ takes the block form (where $\bar{C}^i = C^i/(J-1)$):

$$T^i = \begin{pmatrix} H^i & \bar{C}^i & \bar{C}^i & ... & \bar{C}^i \\ \bar{C}^i & H^i & \bar{C}^i & ... & \bar{C}^i \\ \bar{C}^i & \bar{C}^i & H^i & ... & \bar{C}^i \\ \vdots & \vdots & \vdots & \ddots & \vdots \\ \bar{C}^i & \bar{C}^i & \bar{C}^i & ... & H^i \end{pmatrix},$$

A key point is that $T$ is *dynamically sized* since the number of agents may vary. This motivates the the normalizing factor $J-1$ in equation (2), which rescales the communication vector by the number of communicating agents. Note also that $T^i$ is permutation invariant, thus the order of the agents does not matter.

At the first layer of the model an encoder function $h_j^0 = r(s_j)$ is used. This takes as input state-view $s_j$ and outputs feature vector $h_j^0$ (in $\mathbb{R}^{d_0}$ for some $d_0$). The form of the encoder is problem dependent, but for most of our tasks it is a single layer neural network. Unless otherwise noted, $c_j^0 = 0$ for all $j$. At the output of the model, a decoder function $q(h_j^K)$ is used to output a distribution over the space of actions. $q(.)$ takes the form of a single layer network, followed by a softmax. To produce a discrete action, we sample from this distribution: $a_j \sim q(h_j^K)$.

Thus the entire model (shown in Fig. 1), which we call a Communication Neural Net (CommNet), (i) takes the state-view of all agents $\mathbf{s}$, passes it through the encoder $\mathbf{h}^0 = r(\mathbf{s})$, (ii) iterates $\mathbf{h}$ and $\mathbf{c}$ in equations (1) and (2) to obtain $\mathbf{h}^K$, (iii) samples actions $\mathbf{a}$ for all agents, according to $q(\mathbf{h}^K)$.

### 2.2 Model Extensions

**Local Connectivity:** An alternative to the broadcast framework described above is to allow agents to communicate to others within a certain range. Let $N(j)$ be the set of agents present within

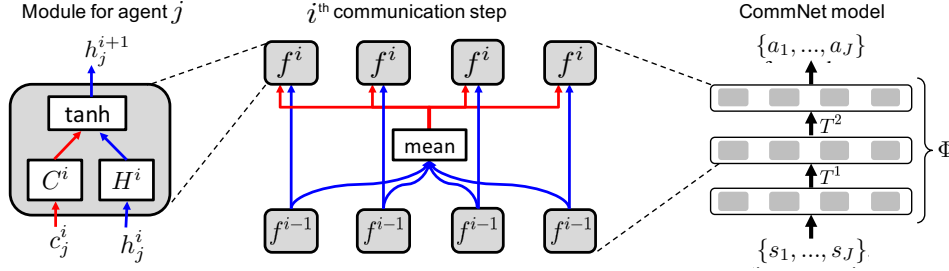

Figure 1: An overview of our CommNet model. Left: view of module $f^i$ for a single agent $j$. Note that the parameters are shared across all agents. Middle: a single communication step, where each agents modules propagate their internal state $h$, as well as broadcasting a communication vector $c$ on a common channel (shown in red). Right: full model $\Phi$, showing input states $s$ for each agent, two communication steps and the output actions for each agent.

communication range of agent $j$. Then (2) becomes:

$$c_j^{i+1} = \frac{1}{|N(j)|} \sum_{j' \in N(j)} h_{j'}^{i+1}.$$ 

(3)

As the agents move, enter and exit the environment, $N(j)$ will change over time. In this setting, our model has a natural interpretation as a dynamic graph, with $N(j)$ being the set of vertices connected to vertex $j$ at the current time. The edges within the graph represent the communication channel between agents, with (3) being equivalent to belief propagation [22]. Furthermore, the use of multi-layer nets at each vertex makes our model similar to an instantiation of the GGSNN work of Li *et al.* [14].

**Skip Connections:** For some tasks, it is useful to have the input encoding $h_j^0$ present as an input for communication steps beyond the first layer. Thus for agent $j$ at step $i$, we have:

$$h_j^{i+1} = f^i(h_j^i, c_j^i, h_j^0).$$ 

(4)

**Temporal Recurrence:** We also explore having the network be a recurrent neural network (RNN). This is achieved by simply replacing the communication step $i$ in Eqn. (1) and (2) by a time step $t$, and using the same module $f^t$ for all $t$. At every time step, actions will be sampled from $q(h_j^t)$. Note that agents can leave or join the swarm at any time step. If $f^t$ is a single layer network, we obtain plain RNNs that communicate with each other. In later experiments, we also use an LSTM as an $f^t$ module.

## 3 Related Work

Our model combines a deep network with reinforcement learning [8, 20, 13]. Several recent works have applied these methods to multi-agent domains, such as Go [16, 24] and Atari games [29], but they assume full visibility of the environment and lack communication. There is a rich literature on multi-agent reinforcement learning (MARL) [1], particularly in the robotics domain [18, 25, 5, 21, 2]. Amongst fully cooperative algorithms, many approaches [12, 15, 33] avoid the need for communication by making strong assumptions about visibility of other agents and the environment. Others use communication, but with a pre-determined protocol [30, 19, 37, 17].

A few notable approaches involve learning to communicate between agents under partial visibility: Kasai *et al.* [10] and Varshavskaya *et al.* [32], both use distributed tabular-RL approaches for simulated tasks. Giles & Jim [6] use an evolutionary algorithm, rather than reinforcement learning. Guestrin *et al.* [7] use a single large MDP to control a collection of agents, via a factored message passing framework where the messages are learned. In contrast to these approaches, our model uses a deep network for both agent control and communication.

From a MARL perspective, the closest approach to ours is the concurrent work of Foerster *et al.* [4]. This also uses a deep reinforcement learning in multi-agent partially observable tasks, specifically two riddle problems (similar in spirit to our *levers* task) which necessitate multi-agent communication.

Like our approach, the communication is learned rather than being pre-determined. However, the agents communicate in a discrete manner through their actions. This contrasts with our model where multiple continuous communication cycles are used at each time step to decide the actions of all agents. Furthermore, our approach is amenable to dynamic variation in the number of agents.

The Neural GPU [9] has similarities to our model but differs in that a 1-D ordering on the input is assumed and it employs convolution, as opposed to the global pooling in our approach (thus permitting unstructured inputs). Our model can be regarded as an instantiation of the GNN construction of Scarselli *et al.* [23], as expanded on by Li *et al.* [14]. In particular, in [23], the output of the model is the fixed point of iterating equations (3) and (1) to convergence, using recurrent models. In [14], these recurrence equations are unrolled a fixed number of steps and the model trained via backprop through time. In this work, we do not require the model to be recurrent, neither do we aim to reach steady state. Additionally, we regard Eqn. (3) as a pooling operation, conceptually making our model a single feed-forward network with local connections.

# 4 Experiments

## 4.1 Baselines

We describe three baselines models for $\Phi$ to compare against our model.

**Independent controller:** A simple baseline is where agents are controlled independently without any communication between them. We can write $\Phi$ as $\mathbf{a} = \{\phi(s_1), ..., \phi(s_J)\}$, where $\phi$ is a per-agent controller applied independently. The advantages of this communication-free model is modularity and flexibility[1]. Thus it can deal well with agents joining and leaving the group, but it is not able to coordinate agents' actions.

**Fully-connected:** Another obvious choice is to make $\Phi$ a fully-connected multi-layer neural network, that takes concatenation of $h_j^0$ as an input and outputs actions $\{a_1, ..., a_J\}$ using multiple output softmax heads. It is equivalent to allowing $T$ to be an arbitrary matrix with fixed size. This model would allow agents to communicate with each other and share views of the environment. Unlike our model, however, it is not modular, inflexible with respect to the composition and number of agents it controls, and even the order of the agents must be fixed.

**Discrete communication:** An alternate way for agents to communicate is via discrete symbols, with the meaning of these symbols being learned during training. Since $\Phi$ now contains discrete operations and is not differentiable, reinforcement learning is used to train in this setting. However, unlike actions in the environment, an agent has to output a discrete symbol at every communication step. But if these are viewed as *internal* time steps of the agent, then the communication output can be treated as an action of the agent at a given (internal) time step and we can directly employ policy gradient [35].

At communication step $i$, agent $j$ will output the index $w_j^i$ corresponding to a particular symbol, sampled according to:

$$w_j^i \sim \text{Softmax}(Dh_j^i) \tag{5}$$

where matrix $D$ is the model parameter. Let $\hat{w}$ be a 1-hot binary vector representation of $w$. In our broadcast framework, at the next step the agent receives a bag of vectors from all the other agents (where $\wedge$ is the element-wise OR operation):

$$c_j^{i+1} = \bigwedge_{j' \neq j} \hat{w}_{j'}^i \tag{6}$$

## 4.2 Simple Demonstration with a Lever Pulling Task

We start with a very simple game that requires the agents to communicate in order to win. This consists of $m$ levers and a pool of $N$ agents. At each round, $m$ agents are drawn at random from the total pool of $N$ agents and they must each choose a lever to pull, simultaneously with the other $m-1$ agents, after which the round ends. The goal is for each of them to pull a *different* lever. Correspondingly, all agents receive reward proportional to the number of distinct levers pulled. Each agent can see its own identity, and nothing else, thus $s_j = j$.

We implement the game with $m = 5$ and $N = 500$. We use a CommNet with two communication steps ($K = 2$) and skip connections from (4). The encoder $r$ is a lookup-table with $N$ entries of 128D. Each $f^i$ is a two layer neural net with ReLU non-linearities that takes in the concatenation of $(h^i, c^i, h^0)$, and outputs a 128D vector. The decoder is a linear layer plus softmax, producing a distribution over the $m$ levers, from which we sample to determine the lever to be pulled. We compare it against the independent controller, which has the same architecture as our model except that communication $c$ is zeroed. The results are shown in Table 1. The metric is the number of distinct levers pulled divided by $m = 5$, averaged over 500 trials, after seeing 50000 batches of size 64 during training. We explore both reinforcement (see the supplementary material) and direct supervision (using the solution given by sorting the agent IDs, and having each agent pull the lever according to its relative order in the current $m$ agents). In both cases, the CommNet performs significantly better than the independent controller. See the supplementary material for an analysis of a trained model.

| Model Φ | Training method | |
|---|---|---|
| | Supervised | Reinforcement |
| Independent | 0.59 | 0.59 |
| CommNet | **0.99** | **0.94** |

Table 1: Results of lever game (#distinct levers pulled)/(#levers) for our CommNet and independent controller models, using two different training approaches. Allowing the agents to communicate enables them to succeed at the task.

## 4.3 Multi-turn Games

In this section, we consider two multi-agent tasks using the MazeBase environment [26] that use reward as their training signal. The first task is to control cars passing through a traffic junction to maximize the flow while minimizing collisions. The second task is to control multiple agents in combat against enemy bots.

We experimented with several module types. With a feedforward MLP, the module $f^i$ is a single layer network and $K = 2$ communication steps are used. For an RNN module, we also used a single layer network for $f^t$, but shared parameters across time steps. Finally, we used an LSTM for $f^t$. In all modules, the hidden layer size is set to 50. MLP modules use skip-connections. Both tasks are trained for 300 epochs, each epoch being 100 weight updates with RMSProp [31] on mini-batch of 288 game episodes (distributed over multiple CPU cores). In total, the models experience $\sim$8.6M episodes during training. We repeat all experiments 5 times with different random initializations, and report mean value along with standard deviation. The training time varies from a few hours to a few days depending on task and module type.

### 4.3.1 Traffic Junction

This consists of a 4-way junction on a $14 \times 14$ grid as shown in Fig. 2(left). At each time step, new cars enter the grid with probability $p_{\text{arrive}}$ from each of the four directions. However, the total number of cars at any given time is limited to $N_{\text{max}} = 10$. Each car occupies a single cell at any given time and is randomly assigned to one of three possible routes (keeping to the right-hand side of the road). At every time step, a car has two possible actions: *gas* which advances it by one cell on its route or *brake* to stay at its current location. A car will be removed once it reaches its destination at the edge of the grid.

Two cars *collide* if their locations overlap. A collision incurs a reward $r_{coll} = -10$, but does not affect the simulation in any other way. To discourage a traffic jam, each car gets reward of $\tau r_{time} = -0.01\tau$ at every time step, where $\tau$ is the number time steps passed since the car arrived. Therefore, the total reward at time $t$ is:

$$r(t) = C^t r_{coll} + \sum_{i=1}^{N^t} \tau_i r_{time},$$

where $C^t$ is the number of collisions occurring at time $t$, and $N^t$ is number of cars present. The simulation is terminated after 40 steps and is classified as a failure if one or more more collisions have occurred.

Each car is represented by one-hot binary vector set $\{n, l, r\}$, that encodes its unique ID, current location and assigned route number respectively. Each agent controlling a car can only observe other cars in its vision range (a surrounding $3 \times 3$ neighborhood), but it can communicate to all other cars.

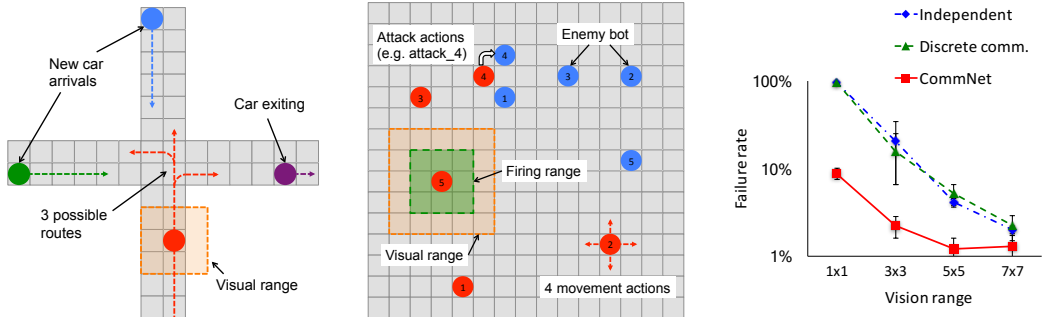

Figure 2: Left: Traffic junction task where agent-controlled cars (colored circles) have to pass the through the junction without colliding. Middle: The combat task, where model controlled agents (red circles) fight against enemy bots (blue circles). In both tasks each agent has limited visibility (orange region), thus is not able to see the location of all other agents. Right: As visibility in the environment decreases, the importance of communication grows in the traffic junction task.

The state vector $s_j$ for each agent is thus a concatenation of all these vectors, having dimension $3^2 \times |n| \times |l| \times |r|$.

In Table 2(left), we show the probability of failure of a variety of different model $\Phi$ and module $f$ pairs. Compared to the baseline models, CommNet significantly reduces the failure rate for all module types, achieving the best performance with LSTM module (a video showing this model before and after training can be found at `http://cims.nyu.edu/~sainbar/commnet`).

We also explored how partial visibility within the environment effects the advantage given by communication. As the vision range of each agent decreases, the advantage of communication increases as shown in Fig. 2(right). Impressively, with zero visibility (the cars are driving blind) the CommNet model is still able to succeed 90% of the time.

Table 2(right) shows the results on easy and hard versions of the game. The easy version is a junction of two one-way roads, while the harder version consists from four connected junctions of two-way roads. Details of the other game variations can be found in the supplementary material. Discrete communication works well on the easy version, but the CommNet with local connectivity gives the best performance on the hard case.

### 4.3.2 Analysis of Communication

We now attempt to understand what the agents communicate when performing the junction task. We start by recording the hidden state $h_j^i$ of each agent and the corresponding *communication vectors* $\tilde{c}_j^{i+1} = C^{i+1} h_j^i$ (the contribution agent $j$ at step $i + 1$ makes to the hidden state of other agents). Fig. 3(left) and Fig. 3(right) show the 2D PCA projections of the communication and hidden state vectors respectively. These plots show a diverse range of hidden states but far more clustered communication vectors, many of which are close to zero. This suggests that while the hidden state carries information, the agent often prefers not to communicate it to the others unless necessary. This is a possible consequence of the broadcast channel: if everyone talks at the same time, no-one can understand. See the supplementary material for norm of communication vectors and brake locations.

| Model $\Phi$ | Module $f()$ type | | |
| --- | --- | --- | --- |
| | MLP | RNN | LSTM |
| Independent | 20.6± 14.1 | 19.5± 4.5 | 9.4± 5.6 |
| Fully-connected | 12.5± 4.4 | 34.8± 19.7 | 4.8± 2.4 |
| Discrete comm. | 15.8± 9.3 | 15.2± 2.1 | 8.4± 3.4 |
| CommNet | **2.2**± **0.6** | **7.6**± **1.4** | **1.6**± **1.0** |

| Model $\Phi$ | Other game versions | |
| --- | --- | --- |
| | Easy (MLP) | Hard (RNN) |
| Independent | 15.8± 12.5 | 26.9± 6.0 |
| Discrete comm. | 1.1± 2.4 | 28.2± 5.7 |
| CommNet | **0.3**± **0.1** | 22.5± 6.1 |
| CommNet local | - | **21.1**± **3.4** |

Table 2: Traffic junction task. Left: failure rates (%) for different types of model and module function $f(.)$. CommNet consistently improves performance, over the baseline models. Right: Game variants. In the easy case, discrete communication does help, but still less than CommNet. On the hard version, local communication (see Section 2.2) does at least as well as broadcasting to all agents.

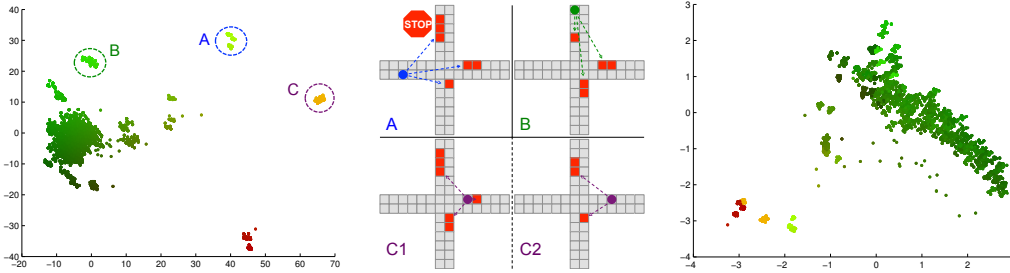

Figure 3: Left: First two principal components of communication vectors $\tilde{c}$ from multiple runs on the traffic junction task Fig. 2(left). While the majority are "silent" (i.e. have a small norm), distinct clusters are also present. Middle: for three of these clusters, we probe the model to understand their meaning (see text for details). Right: First two principal components of hidden state vectors $h$ from the same runs as on the left, with corresponding color coding. Note how many of the "silent" communication vectors accompany non-zero hidden state vectors. This shows that the two pathways carry different information.

To better understand the meaning behind the communication vectors, we ran the simulation with only two cars and recorded their communication vectors and locations whenever one of them braked. Vectors belonging to the clusters A, B & C in Fig. 3(left) were consistently emitted when one of the cars was in a specific location, shown by the colored circles in Fig. 3(middle) (or pair of locations for cluster C). They also strongly correlated with the other car braking at the locations indicated in red, which happen to be relevant to avoiding collision.

### 4.3.3 Combat Task

We simulate a simple battle involving two opposing teams in a $15 \times 15$ grid as shown in Fig. 2(middle). Each team consists of $m = 5$ agents and their initial positions are sampled uniformly in a $5 \times 5$ square around the team center, which is picked uniformly in the grid. At each time step, an agent can perform one of the following actions: move one cell in one of four directions; attack another agent by specifying its ID $j$ (there are $m$ attack actions, each corresponding to one enemy agent); or do nothing. If agent A attacks agent B, then B's health point will be reduced by 1, but only if B is inside the firing range of A (its surrounding $3 \times 3$ area). Agents need one time step of cooling down after an attack, during which they cannot attack. All agents start with 3 health points, and die when their health reaches 0. A team will win if all agents in the other team die. The simulation ends when one team wins, or neither of teams win within 40 time steps (a draw).

The model controls one team during training, and the other team consist of bots that follow a hard-coded policy. The bot policy is to attack the nearest enemy agent if it is within its firing range. If not, it approaches the nearest visible enemy agent within visual range. An agent is visible to all bots if it is inside the visual range of any individual bot. This shared vision gives an advantage to the bot team. When input to a model, each agent is represented by a set of one-hot binary vectors $\{i, t, l, h, c\}$ encoding its unique ID, team ID, location, health points and cooldown. A model controlling an agent also sees other agents in its visual range ($3 \times 3$ surrounding area). The model gets reward of -1 if the team loses or draws at the end of the game. In addition, it also get reward of $-0.1$ times the total health points of the enemy team, which encourages it to attack enemy bots.

| Model $\Phi$ | Module $f()$ type | | |
|---|---|---|---|
| | MLP | RNN | LSTM |
| Independent | $34.2\pm 1.3$ | $37.3\pm 4.6$ | $44.3\pm 0.4$ |
| Fully-connected | $17.7\pm 7.1$ | $2.9\pm 1.8$ | $19.6\pm 4.2$ |
| Discrete comm. | $29.1\pm 6.7$ | $33.4\pm 9.4$ | $46.4\pm 0.7$ |
| CommNet | $\mathbf{44.5}\pm \mathbf{13.4}$ | $\mathbf{44.4}\pm \mathbf{11.9}$ | $\mathbf{49.5}\pm \mathbf{12.6}$ |

| Model $\Phi$ | Other game variations (MLP) | | |
|---|---|---|---|
| | $m = 3$ | $m = 10$ | $5 \times 5$ vision |
| Independent | $29.2\pm 5.9$ | $30.5\pm 8.7$ | $60.5\pm 2.1$ |
| CommNet | $\mathbf{51.0}\pm \mathbf{14.1}$ | $\mathbf{45.4}\pm \mathbf{12.4}$ | $\mathbf{73.0}\pm \mathbf{0.7}$ |

Table 3: Win rates (%) on the combat task for different communication approaches and module choices. Continuous consistently outperforms the other approaches. The fully-connected baseline does worse than the independent model without communication. On the right we explore the effect of varying the number of agents $m$ and agent visibility. Even with 10 agents on each team, communication clearly helps.

Table 3 shows the win rate of different module choices with various types of model. Among different modules, the LSTM achieved the best performance. Continuous communication with CommNet improved all module types. Relative to the independent controller, the fully-connected model degraded performance, but the discrete communication improved LSTM module type. We also explored several variations of the task: varying the number of agents in each team by setting $m = 3, 10$, and increasing visual range of agents to $5 \times 5$ area. The result on those tasks are shown on the right side of Table 3. Using CommNet model consistently improves the win rate, even with the greater environment observability of the $5 \times 5$ vision case.

### 4.4 bAbI Tasks

We apply our model to the bAbI [34] toy Q & A dataset, which consists of 20 tasks each requiring different kind of reasoning. The goal is to answer a question after reading a short story. We can formulate this as a multi-agent task by giving each sentence of the story its own agent. Communication among agents allows them to exchange useful information necessary to answer the question.

The input is $\{s_1, s_2, ..., s_J, q\}$, where $s_j$ is $j$'th sentence of the story, and $q$ is the question sentence. We use the same encoder representation as [27] to convert them to vectors. The $f(.)$ module consists of a two-layer MLP with ReLU non-linearities. After $K = 2$ communication steps, we add the final hidden states together and pass it through a softmax decoder layer to sample an output word $y$. The model is trained in a supervised fashion using a cross-entropy loss between $y$ and the correct answer $y^*$. The hidden layer size is set to 100 and weights are initialized from $N(0, 0.2)$. We train the model for 100 epochs with learning rate 0.003 and mini-batch size 32 with Adam optimizer [11] ($\beta_1 = 0.9, \beta_2 = 0.99, \epsilon = 10^{-6}$). We used 10% of training data as validation set to find optimal hyper-parameters for the model.

Results on the 10K version of the bAbI task are shown in Table 4, along with other baselines (see the supplementary material for a detailed breakdown). Our model outperforms the LSTM baseline, but is worse than the MemN2N model [27], which is specifically designed to solve reasoning over long stories. However, it successfully solves most of the tasks, including ones that require information sharing between two or more agents through communication.

|  | Mean error (%) | Failed tasks (err. > 5%) |
|---|---|---|
| LSTM [27] | 36.4 | 16 |
| MemN2N [27] | 4.2 | 3 |
| DMN+ [36] | **2.8** | **1** |
| Independent (MLP module) | 15.2 | 9 |
| CommNet (MLP module) | 7.1 | 3 |

Table 4: Experimental results on bAbI tasks.

## 5 Discussion and Future Work

We have introduced CommNet, a simple controller for MARL that is able to learn continuous communication between a dynamically changing set of agents. Evaluations on four diverse tasks clearly show the model outperforms models without communication, fully-connected models, and models using discrete communication. Despite the simplicity of the broadcast channel, examination of the traffic task reveals the model to have learned a sparse communication protocol that conveys meaningful information between agents. Code for our model (and baselines) can be found at `http://cims.nyu.edu/~sainbar/commnet/`.

One aspect of our model that we did not fully exploit is its ability to handle heterogenous agent types and we hope to explore this in future work. Furthermore, we believe the model will scale gracefully to large numbers of agents, perhaps requiring more sophisticated connectivity structures; we also leave this to future work.

### Acknowledgements

The authors wish to thank Daniel Lee and Y-Lan Boureau for their advice and guidance. Rob Fergus is grateful for the support of CIFAR.

## Footnotes

[1] Assuming $s_j$ includes the identity of agent $j$.

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
