[Supplementary Material]

# Learning Multiagent Communication with Backpropagation Supplementary Material

**Sainbayar Sukhbaatar**
Dept. of Computer Science
Courant Institute, New York University
sainbar@cs.nyu.edu

**Arthur Szlam**
Facebook AI Research
New York
aszlam@fb.com

**Rob Fergus**
Facebook AI Research
New York
robfergus@fb.com

## 1 Reinforcement Training

We use policy gradient [5] with a state specific baseline for delivering a gradient to the model. Denote the states in an episode by $s(1), ..., s(T)$, and the actions taken at each of those states as $a(1), ..., a(T)$, where $T$ is the length of the episode. The baseline is a scalar function of the states $b(s, \theta)$, computed via an extra head on the model producing the action probabilities. Beside maximizing the expected reward with policy gradient, the models are also trained to minimize the distance between the baseline value and actual reward. Thus after finishing an episode, we update the model parameters $\theta$ by

$$\Delta\theta = \sum_{t=1}^{T} \left[ \frac{\partial \log p(a(t)|s(t), \theta)}{\partial \theta} \left( \sum_{i=t}^{T} r(i) - b(s(t), \theta) \right) - \alpha \frac{\partial}{\partial \theta} \left( \sum_{i=t}^{T} r(i) - b(s(t), \theta) \right)^2 \right].$$

(1)

Here $r(t)$ is reward given at time $t$, and the hyperparameter $\alpha$ is for balancing the reward and the baseline objectives, which set to 0.03 in all experiments.

## 2 Lever Pulling Task Analysis

Figure 1: 3D PCA plot of hidden states of agents

Here we analyze a CommNet model trained with supervision on the lever pulling task. The supervision uses the sorted ordering of agent IDs to assign target actions. For each agent, we concatenate its

hidden layer activations during game playing. Fig. 1 shows 3D PCA plot of those vectors, where color intensity represents agent's ID. The smooth ordering suggests that agents are communicating their IDs, enabling them to solve the task.

## 3 Details of Traffic Junction

We use curriculum learning [1] to make the training easier. In first 100 epochs of training, we set $p_{\text{arrive}} = 0.05$, but linearly increased it to 0.2 during next 100 epochs. Finally, training continues for another 100 epochs. The learning rate is fixed at 0.003 throughout. We also implemented additional easy and hard versions of the game, the latter being shown in Fig.2.

The easy version is a junction of two one-way roads on a $7 \times 7$ grid. There are two arrival points, each with two possible routes. During curriculum, we increase $N_{\text{total}}$ from 3 to 5, and $p_{\text{arrive}}$ from 0.1 to 0.3.

The harder version consists from four connected junctions of two-way roads in $18 \times 18$ as shown in Fig. 2. There are 8 arrival points and 7 different routes for each arrival point. We set $N_{\text{total}} = 20$, and increased $p_{\text{arrive}}$ from 0.02 to 0.05 during curriculum.

Figure 2: A harder version of traffic task with four connected junctions.

## 4 Traffic Junction Analysis

Here we visualize the average norm of the communication vectors in Fig. 3(left) and brake locations over the $14 \times 14$ spatial grid in Fig. 3(right). In each of the four incoming directions, there is one location where communication signal is stronger. The brake pattern shows that cars coming from left never yield to other directions.

Figure 3: (left) Average norm of communication vectors (right) Brake locations

# 5   bAbI Tasks Details

Here we give further details of the model setup and training, as well as a breakdown of results in Table 1.

Let the task be $\{s_1, s_2, ..., s_J, q, y^*\}$, where $s_j$ is $j$'th sentence of story, $q$ is the question sentence and $y^*$ is the correct answer word (when answer is multiple words, we simply concatenate them into single word). Then the input to the model is

$$h_j^0 = r(s_j, \theta_0), \quad c_j^0 = r(q, \theta_q).$$

Here, we use simple position encoding [4] as $r$ to convert sentences into fixed size vectors. Also, the initial communication is used to broadcast the question to all agents. Since the temporal ordering of sentences is relevant in some tasks, we add special temporal word "$t = J - j$" to $s_j$ for all $j$.

For $f$ module, we use a 2 layer network with skip connection, that is

$$h_j^{i+1} = \sigma(W_i \sigma(H^i h_j^i + C^i c_j^i + h_j^0)),$$

where $\sigma$ is ReLU non-linearity (bias terms are omitted for clarity). After $K = 2$ communication steps, the model outputs an answer word by

$$y = Softmax(D \sum_{j=1}^{J} h_j^K)$$

Since we have the correct answer during training, we will do supervised learning by using cross entropy cost on $\{y^*, y\}$. The hidden layer size is set 100 and weights are initialized from $N(0, 0.2)$. We train the model 100 epochs with learning rate 0.003 and mini-batch size 32 with Adam optimizer [2] ($\beta_1 = 0.9, \beta_2 = 0.99, \epsilon = 10^{-6}$). We used 10% of training data as validation set to find optimal hyper-parameters for the model.

| | Error on tasks (%) | | | | | | | Mean error | Failed tasks |
|---|---|---|---|---|---|---|---|---|---|
| | 2 | 3 | 15 | 16 | 17 | 18 | 19 | (%) | (err. > 5%) |
| LSTM [4] | 81.9 | 83.1 | 78.7 | 51.9 | 50.1 | 6.8 | 90.3 | 36.4 | 16 |
| MemN2N [4] | **0.3** | 2.1 | **0.0** | 51.8 | 18.6 | 5.3 | 2.3 | 4.2 | 3 |
| DMN+ [6] | **0.3** | **1.1** | **0.0** | 45.3 | 4.2 | 2.1 | **0.0** | **2.8** | **1** |
| Neural Reasoner+ [3] | - | - | - | - | **0.9** | - | 1.6 | - | - |
| Independent (MLP module) | 69.0 | 69.5 | 29.4 | 47.4 | 4.0 | **0.6** | 45.8 | 15.2 | 9 |
| CommNet (MLP module) | 3.2 | 68.3 | **0.0** | 51.3 | 15.1 | 1.4 | **0.0** | 7.1 | 3 |

Table 1: Experimental results on bAbI tasks. Only showing some of the task with high errors.