[Reviews · NeurIPS 2016]

Reviewer 1

Summary

This paper proposed a neural network model for communication between agents in a cooperative reinforcement learning problem, where the agents learn the communication along with the policies. Each agent has a neural network that takes the previous hidden activations and the communication signals from other agents as its inputs. The neural networks have shared weights so that the entire multi-agent system can be dynamically sized and is order invariant. The model is learned via policy gradient and back propagation.

Qualitative Assessment

The approach learns the “language” for communication between different agents, which is an interesting progress. The difference between this work and prior work on learning to communicate between different agents is the incorporation of deep nets into the multi-agent reinforcement learning framework. My main concern about this work is that the approach requires synchronization during the communication process. That is, in order to compute the hidden activations for the next layer, it requires all the agents to receive communication signals from the other agents. This would be an important restriction in practice. A typo on line 91 of page 3: “from the this distribution”.

Confidence in this Review

2-Confident (read it all; understood it all reasonably well)


Reviewer 2

Summary

This paper introduces a neural model for learning multi-agent fully cooperative tasks. In such a task, many agents cooperate (getting the same reward) to achieve a task. These agents have their own neural controllers but can also communicate. The paper studies different communication topologies in both discrete and continuous setting and shows promising results, esp. in the continuous communitation setting (where agents can send vectors of floats).

Qualitative Assessment

In the spirit of The Society of Mind, the paper explores how to train a number of agents to coordinate to achieve a task. It is impressive that this can be achieved -- and it looks from the results that introducing a continuous communication channel and back-propagating through the whole system are crucial to the success. What is crucial to the success, though, also looks like exactly the factors that would limit the usefulness of this approach. For all tasks except bAbI, the authors provide no baseline other than their own system. This is strange -- please, provide a baseline from a centralized controller that has access to all observations of all agents and makes a central decision for all of them. Would it be superior? If a centralized controller is superior, why is it useful to have many agents? There could be many reasons, but it seems that they rely on new agents -- randomly initialized -- being able to join the group and be trained together with the existing ones. Would that work in the presented framework? (Agents can join already, but can they be re-trained?) In all cases, a baseline with centralized controller should be presented for the cooperative games tasks.

Confidence in this Review

2-Confident (read it all; understood it all reasonably well)


Reviewer 3

Summary

The paper presents a model that learns to communicate among a variable number of agents all trying to perform a joint task together and being rewarded by reinforcement learning. Agents do not directly communicate with each other but instead all participate in a joint broadcasting. Experiments on several artificial tasks are presented comparing various versions of the model.

Qualitative Assessment

The proposed model is interesting because it uses backprop over each agent to help each of them participate in a joint objective. The architecture is flexible and can accept a variable number of agents at any given time during the optimization process, which is neat. I have a few concerns/questions as follows: - Why is equation 1 expressed as an update of the parameters rather than as a loss? it would be clearer if we could see what is the actual joint loss optimized by the model. - I am not an expert in multi-agent reinforcement learning, but I was surprised that the proposed model was not compared to any previous approach; instead, multiple variants of the model are compared in the experiments. This also makes it hard for the reader to assess how difficult the chosen artificial tasks really were. ========== After reading the authors' response, I'm bumping up my scores as they have answered my concerns.

Confidence in this Review

3-Expert (read the paper in detail, know the area, quite certain of my opinion)


Reviewer 4

Summary

The paper deals with centralized learning in the case of multi-agent collaboration problems. More specifically, it proposes a deep-learning approach for learning to collaborate when every agent needs to communicate with others (each agent only gets a partial view of the environment), but no communication protocol has been stated beforehand.

Qualitative Assessment

The model is a deep network which consists of a stack of layers, with parameter sharing between modules of a same layer. This parameter sharing allows the number of agents to vary during the task. Also, it allows to drastically reduce the number of parameters to be learned. The key idea of the paper is to use the output of every module of a given layer to build the communication input for the next layer. While this appears to obtain interesting results in the reported experiments, I find this proposal very straightforward and poorly innovative, as it corresponds to a quiet classical neural network structure. In fact, that is a classical network with shared parameters, except that the output of any agent $i$ is concatenated with (rather than added to) the summed outputs of every others to form the input of this agent $i$ for the next layer. I am not sure about the intuition of what this could bring in the context of the considered tasks. Experiments should have at least compared the proposal with a classical network with parameter sharing to highlight the benefits of the proposed mechanism. Also, a centralized controller that has access to all observations of all agents should perform better than the studied setting. So, if the dense baseline corresponds to such a centralized controller scheme (as suggested by the authors answer), there is no reason for me why this baseline does not get greatly better results... Since getting the whole information, this baseline should perform better than approaches with communication between partial views. Maybe that is because it has too many parameters and this is due to an optimization problem ? That's also why I asked a baseline as dense but with shared parameters between modules of a same layer to be able to compare with the proposal and to lower the learning complexity. Anyway, I am not sure on how the dense baseline works in the various settings... Other comments: - As the main argument is to maintain a network where the structure can be dynamically sized, it would have been useful to give in introduction some examples of applications where this dynamicity is particularly important - I am not sure about the relevance of presenting the reinforcement policy gradient in section 2, as : 1) the proposed model can be learned via supervised learning in some applications and 2) this is not at the heart of the proposal at all. I would give it (with more details to be more self-contained) after the presentation of the model (maybe in a section Learning or something like that). Like this it confuses the discourse a lot from my point of view - Not enough details of the experimental settings are given in the main paper. Very important points such as inputs given to the agents are given in the supplementary material. This complicates the analysis of the results - Not enough explanations / intuitions are given w.r.t. what agents communicate for the different tasks. Wouldn't it be possible to give a simple example of what is learned for instance in a tiny lever pulling instance ? The section 5.2.2 does not help a lot to understand what is exchanged between agents

Confidence in this Review

2-Confident (read it all; understood it all reasonably well)